# A Unified Cortical Circuit Model with Divisive Normalization and Self-Excitation for Robust Representation and Memory Maintenance

## Abstract

Robust information representation and its persistent maintenance are fundamental for higher cognitive functions. Existing models employ distinct neural mechanisms to separately address noise-resistant processing or information maintenance, yet a unified framework integrating both operations remains elusive—a critical gap in understanding cortical computation. Here, we introduce a recurrent neural circuit that combines divisive normalization with self-excitation to achieve both robust encoding and stable retention of normalized inputs. Mathematical analysis shows that, for suitable parameter regimes, the system forms a continuous attractor with two key properties: (1) *input-proportional stabilization* during stimulus presentation; and (2) *self-sustained memory states* persisting after stimulus offset. We demonstrate the model's versatility in two canonical tasks: (a) noise-robust encoding in a random-dot kinematogram (RDK) paradigm; and (b) approximate Bayesian belief updating in a probabilistic Wisconsin Card Sorting Test (pWCST). This work establishes a unified mathematical framework that bridges noise suppression, working memory, and approximate Bayesian inference within a single cortical microcircuit, offering fresh insights into the brain's canonical computation and guiding the design of biologically plausible artificial neural architectures.

## 1 Introduction

Biological intelligence has garnered widespread attention due to its ability to efficiently and reliably process information in complex, dynamic, and uncertain environments. This capability enables the performance of intricate cognitive tasks, allowing for flexible and efficient adaptation to changing circumstances. In order to achieve a reliable understanding of the external environment and plan subsequent decisions and actions, the brain must effectively perform at least two critical cognitive computations: (i) *Noise-resistant neural coding*, which filters out irrelevant variability and preserves key signal features for further processing, and (ii) *Stable maintenance of information*, which ensures that information is held and represented over time to support memory and planning. For example, in perceptual tasks the brain must denoise noisy sensory inputs to form reliable estimates of motion or contrast (Simoncelli & Heeger, 1998; Deneve et al., 2001), while in working memory and cognitive control it must sustain internal representations of stimuli, rules, or values across delays (Wang, 1999; Compte et al., 2000; Behrens et al., 2007).

Noise suppression in the brain has been most extensively studied in sensory cortices (Simoncelli & Heeger, 1998; Lee & Mumford, 2003; Sawada & Petrov, 2017; Heeger & Zemlianova, 2020; Burg et al., 2021; Ernst et al., 2021; Weiss et al., 2023). While there exists different perspectives, three major mechanisms contribute to noise suppression. Divisive normalization accept that neurons' responses are divided by the activity of a local population (Simoncelli & Heeger, 1998; Sawada & Petrov, 2017; Heeger & Zemlianova, 2020), while Bayesian approach consider the neuron population performing Bayesian inference (Lee & Mumford, 2003; Pouget et al., 2003; Knill & Pouget, 2004; Ma et al., 2006). Besides this, attractor networks are also considered to be noise resistant (Hopfield, 1982; Deneve et al., 2001), making it a candidate model for noise suppression.

On the other hand, persistent maintenance or working memory has been modeled primarily using recurrent attractor networks (Wang, 2002; 2012; Murray et al., 2017; Bouchacourt & Buschman, 2019), or built into network connections through the dynamics of synapses (Compte et al., 2000; Mongillo et al., 2008; Morrison et al., 2008; Wu et al., 2008). Discrete attractors can be used for discrimination or classification, thus also act as decision-making models (Wang, 2002; 2008), while continuous attractors are usually considered as representational manifolds, acting as relatively sensitive working memory models.

Despite these advances, noise-resistant encoding and persistent maintenance have remained largely separate modeling domains: normalization circuits focus on feedforward gain control and noise filtering but fade after input removal Heeger & Zemlianova (2020), whereas attractor models preserve persistent activities yet exhibit limited adaptability, hindering their extension to novel tasks. This dichotomy prompts a pivotal question: by what microcircuit architectures and dynamics does the cerebral cortex reconcile transient noise filtering with flexible, self-sustained representations?

In this paper, we bridge this gap by proposing a Recurrent Divisive Normalization (RDN) circuit model: each excitatory neuron combines its external drive with self-excitation and then divides by a global inhibitory pool. We prove that, under proper parameter conditions, the model not only computes exactly normalized representations of its inputs (robust to noise and gain changes) but also forms a continuous attractor that persistently maintains those representations after input withdrawn. With some conceptual perceptual (random dot kinematogram) and cognitive (probabilistic Wisconsin Card Sorting Test) tasks, we demonstrated the model's versatility. By unifying two canonical mechanisms—divisive normalization and attractor dynamics—this work offers a principled framework for understanding how the brain performs both robust filtering and flexible memory within a single circuit motif, demonstrating the power of this simple model, and could open up new avenues for bio-inspired artificial intelligence.

## 2 A RECURRENT DIVISIVE NORMALIZATION MODEL

We present a firing-rate model that accomplish divisive normalization and self-excitatory recurrent connection. Consider $N$ excitatory units $R_i$ coupled to a single inhibitory pool $G$ (Fig. 1A), where the dynamics of the network is defined by ordinary differential equations:

$$\tau_R \frac{\mathrm{d}R_i}{\mathrm{d}t} = -R_i + \frac{\beta R_i + I_i}{\eta + G}, \quad i = 1, \cdots, N \tag{1}$$

$$\tau_G \frac{\mathrm{d}G}{\mathrm{d}t} = -G + \sum_{i=1}^{N} w_i R_i. \tag{2}$$

Key parameters include: $I_i \geq 0$ (external input to unit $i$), $\beta \geq 0$ (self-excitation strength), $\eta > 0$ (semi-saturation constant), $w_j > 0$ (excitatory-to-inhibitory weights), and $\tau_R, \tau_G > 0$ (time constants).

When $\beta = 0$, the model reduces to the classical divisive normalization (Heeger, 1992; Simoncelli & Heeger, 1998; Keung et al., 2020), and $\beta = 1$ lead to a model similar to some well-known recurrent divisive normalization models Heeger & Mackey (2019); Heeger & Zemlianova (2020); Rawat et al. (2024), making our model as a generalization of divisive normalization model. Despite its structural simplicity, our model exhibits remarkable dynamical properties (Fig. 1). Under appropriate parameter configurations, it forms a continuous attractor networks capable of executing diverse neurocomputational functions, including noise-resistant information processing, persistent memory maintenance, and dynamic learning adaptation.

## 3 MODEL ANALYSIS

### 3.1 STEADY-STATE SOLUTIONS

Setting $\dot{R}_i = \dot{G} = 0$. From Eq. equation 1, we have

$$R_i^*(\eta + G^*) = \beta R_i^* + I_i, \quad R_i^* = \frac{I_i}{\eta - \beta + G^*}.$$

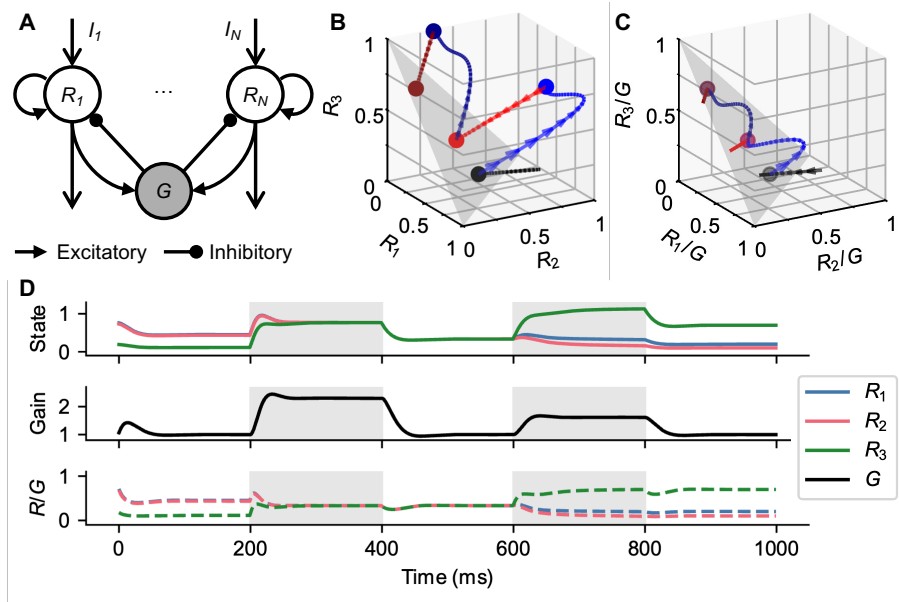

Figure 1: The Recurrent Divisive Normalization (RDN) model and its dynamics. **A.** Schematic of the recurrent divisive normalization circuit; Each excitatory unit $R_i$ combines its external input $I_i$ with a feedback term $\beta R_i$, then divides by $\eta + G$ (Eq. 2). The global pool $G$ aggregates all $R_i$ outputs (Eq. 1). **B–C**: Demonstrating the dynamics of a model ($N = 3$, $\beta = 2$, $\eta = 1$ and $w_i = 1$) with and without input. The model is randomly initialized, and become steady after a while; with a constant input ($I_i = 1$), the model changes its state to a new fixed point where all $R_i$s are the same, and keeps the ratio after input removed. Different input ($I_1 = 0.2, I_2 = 0.1, I_3 = 0.7$) drives the model to a different fixed point. **B.** Attractor manifold and the trajectory of $R_i$ with (blue lines) and without (red lines) input. The colored dots represents the corresponding fixed points. **C.** Attractor manifold and the trajectory of $R_i/G$ with (blue lines) and without (red lines) input; Fixed points align with/without input become the same. **D.** Dynamics of the model, showing state, gain and the readout changing with time during input (shadowed) or not. Blue, pink and green represents the 3 excitatory neurons $R_i$, black represent the inhibitory neuron $G$.

Substituting into Eq. equation 2, let

$$T = \sum_{j=1}^{N} w_j I_j,$$

thus

$$G^* = \sum_{j=1}^{N} w_j R_j = \frac{T}{\eta - \beta + G^*}, \quad G^*(\eta - \beta + G^*) = T.$$

We obtain a quadratic equation of $G^*$:

$$(G^*)^2 + (\eta - \beta)G^* - T = 0. \tag{3}$$

The solution is given by

$$G^* = \frac{-(\eta - \beta) \pm \sqrt{(\eta - \beta)^2 + 4T}}{2}.$$

For $I_i \geq 0$ and $T > 0$, the discriminant $\Delta = (\eta - \beta)^2 + 4T > 0$, and it is easy to obtain that Eq. equation 3 has exactly one root satisfying $G^* > 0$:

$$G^* = \frac{-(\eta - \beta) + \sqrt{(\eta - \beta)^2 + 4T}}{2}, \tag{4}$$

with $\eta - \beta + G^* > 0$ also be satisfied, implying $R_i^* \geq 0$. Therefore, the RDN model has a unique physiologically meaningful steady-state solution $(\boldsymbol{R}^*, G^*) \in \mathbb{R}^{N+1}$.

To analyze the stability of the model, We apply the indirect method of Lyapunov at the fixed point (Strogatz, 2018). Linearize $x = (\delta R_1, \ldots, \delta R_N, \delta G)$. The Jacobian is

$$\boldsymbol{J} = \begin{bmatrix} a\boldsymbol{I}_N & \boldsymbol{b} \\ \boldsymbol{c}^\top & -1/\tau_G \end{bmatrix}, \tag{5}$$

where $\boldsymbol{I}_N$ is an $N$-th order identity matrix, $\boldsymbol{b} = (b_1, \ldots, b_N)^\top$ and $\boldsymbol{c} = (c_1, \ldots, c_N)^\top$,

$$a = \frac{-1 + \beta/(\eta + G^*)}{\tau_R}, \quad b_i = -\frac{\beta R_i^* + I_i}{\tau_R(\eta + G^*)^2}, \quad c_j = \frac{w_j}{\tau_G}.$$

Eigendecomposition of the Jacobian matrix $\boldsymbol{J}$ reveals fixed-point stability (Strogatz, 2018):

1. Eigenvalues with negative real parts indicate contracting dynamics in corresponding state-space dimensions

2. Positive real parts correspond to expanding dynamics

3. Zero real parts reflect marginal stability.

Stable attractors might be useful for integrating noisy information, or for maintaining state or memory.

From Theorem 1 (see Appendix), we know that the eigenvalues of $\boldsymbol{J}$ are

$$\lambda_i = a = \frac{-1 + \beta/(\eta + G^*)}{\tau_R}, \quad (i = 1, \ldots, N-1), \tag{6}$$

$$\lambda_\pm = \frac{(a - 1/\tau_G) \pm \sqrt{(a + 1/\tau_G)^2 - 4G^*/(\eta + G^*)/(\tau_R \tau_G)}}{2}. \tag{7}$$

To analyse the stability of the fixed point analytically is intractable through the above eigenvalues. With simplified parameters $\tau_R = \tau_G = \tau$ and $w_i = 1$ for all $i = 1, \ldots, N$, where the inhibitory pool receives the neurons' inputs equally and have the same time constant with excitatory units, we next analyze the simplified model with the inputs $I_i$ removed from and exerted to it.

### 3.2 Stability of the model without inputs

With the above settings and $I_i \equiv 0$, the dynamics of the model is simplified as

$$\tau \frac{\mathrm{d}R_i}{\mathrm{d}t} = -R_i + \frac{\beta R_i}{\eta + G}, \quad i = 1, \cdots, N \tag{8}$$

$$\tau \frac{\mathrm{d}G}{\mathrm{d}t} = -G + \sum_{j=1}^N R_j, \tag{9}$$

Eq. equation 3 become

$$G^*(\eta - \beta + G^*) = 0.$$

This solves the fixed point $G^* = R_1^* = \cdots = R_N^* = 0$, which is a trivial fixed point, and $G^* = \sum_{i=1}^N R_i^* = \beta - \eta$, which forms an $N - 1$ dimensional continuous attractor manifold (hyperplane) in the $\boldsymbol{R}$ subspace (Fig. 1B).

For the trivial fixed point $G^* = R_1^* = \cdots = R_N^* = 0$, the eigenvalues simplified as

$$\lambda_{1,\cdots,N} = \frac{\beta - \eta}{\tau\eta}, \lambda_{N+1} = -\frac{1}{\tau}. \tag{10}$$

Therefore, the trivial fixed point will be stable and the system will converge if and only if $\beta < \eta$ (Fig. 2, left panel).

For the continuous attractor $G^* = \sum_{i=1}^N R_i^* = \beta - \eta$, eigenvalues $\lambda_{1,\cdots,N-1} = 0$, and

$$\lambda_{N,N+1} = \frac{-1 \pm \sqrt{4\eta/\beta - 3}}{2\tau}. \tag{11}$$

Therefore, the continuous attractor will be stable and the system will converge to the continuous attractor $G^* = \sum_{i=1}^{N} R_i^* = \beta - \eta$ if and only if $\beta > \eta$ (Fig. 2, right panel).

Taken together, the dynamical system governed by Eqs. (8–9) exhibits fundamentally distinct attractor structures in phase space and markedly differentiated dynamical characteristics when the parameters $\beta, \eta$ crossing critical point, yields a transcritical bifurcation in the state space (See Fig. 2 for demonstrations, see also Fig. 1B&D the detailed dynamics with 3 excitatory node ($N = 3$), and the attractor manifold $G^* = \sum_{i=1}^{3} R_i = 1$).

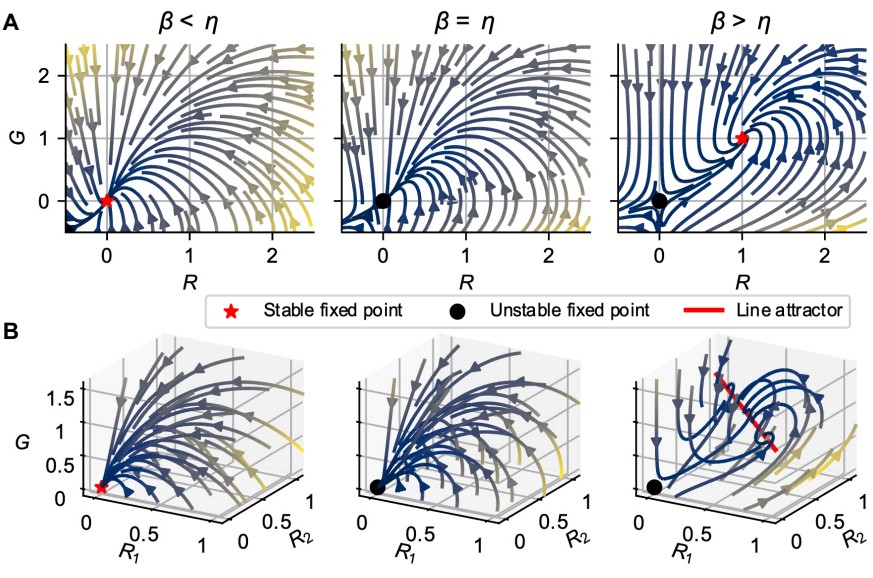

Figure 2: Phase portrait of the model before and after bifurcation (at $\beta = \eta$). **A**. Phase portrait of the simplified model Eqs. (8–9) with one excitatory node ($N = 1$), with $\beta > \eta$, there exist a non-zero stable fixed point $(1, 1)$, which makes our RDN model fundamentally different from the canonical divisive normalization model, enabling persistent memory. **B**. Phase portrait of the simplified RDN model with two excitatory node ($N = 2$), showing the same bifurcation properties with the 1-Node model (A). Number of excitatory neurons can easily extended to fit different kind of tasks, making the RDN model much more useful compared to other models.

Next we discuss the dynamics when the non-zero inputs are *withdrawn* from the system, which is crucial for the working memory capability of the model. Denote $S = \sum_{i=1}^{N} R_i, \rho_i = R_i/S$, which satisfies $\sum_{i=1}^{N} \rho_i = 1$, we can deduce that $\tau d\rho_i/dt = 0$ (see Eq. equation 27, Appendix).

When the system converges to the continuous attractor $G^* = \sum_{i=1}^{N} R_i^* = \beta - \eta$ eventually, the firing rate $R_i$ keeps constant at $\rho_i(\beta - \eta)$, where $\rho_i$ relies on the state before the inputs are withdrawn. In other words, the system maintains the last input-normalized pattern indefinitely, implementing working memory (Fig. 1B&D).

### 3.3 STABILITY OF THE MODEL WITH INPUTS

With inputs $I_i$ exerted to the system, we have

$$\tau \frac{dR_i}{dt} = -R_i + \frac{\beta R_i + I_i}{\eta + G}, \quad i = 1, \cdots, N \tag{12}$$

$$\tau \frac{dG}{dt} = -G + \sum_{j=1}^{N} R_j, \tag{13}$$

Solve as like in Eq. equation 4, we have the fixed point

$$R_i^* = \frac{2I_i}{(\eta - \beta) + \sqrt{(\eta - \beta)^2 + 4T}}, \tag{14}$$

$$G^* = \frac{(\beta - \eta) + \sqrt{(\eta - \beta)^2 + 4T}}{2}. \tag{15}$$

which reveals that the system shifts toward a new fixed point while receiving new inputs. Since

$$\begin{aligned}
\frac{R_i}{G} &= \frac{2I_i}{\sqrt{(\eta - \beta)^2 + 4T} + (\eta - \beta)} \cdot \frac{2}{\sqrt{(\eta - \beta)^2 + 4T} - (\eta - \beta)} \\
&= \frac{4I_i}{(\eta - \beta)^2 + 4T - (\eta - \beta)^2} = \frac{I_i}{T} = \frac{I_i}{\sum_{j=1}^n I_j},
\end{aligned} \tag{16}$$

therefore the system eventually evolves to a fixed point proportional to the input, and one can define a readout rule from the model such that $O_i = R_i/G$, which essentially normalize the inputs (Fig. 1C&D).

Given $I_i > 0$, Eq. equation 7 demonstrates that all eigenvalues satisfy $\lambda_i < 0$ under the condition $\beta < \eta + G^*$. This inequality is inherently guaranteed by Eq. equation 4. Thus the unique steady-state forms an attractor (Theorem 2, see Appendix). One can deduce that the firing rate $R_i$ of each neuron is proportional to the strength of input $I_i$ if the system has converged to the fixed point, following the procedures described in Eq. (28–29). It is also worth noting that with the inputs withdrawn, the firing rate $R_i$ will converge to a steady state $R_i = \rho_i(\eta - \beta)$, where $\rho_i$ relies on the state before the inputs were withdrawn. Therefore, the system stores the memory of the latest inputs, which might be attributed to many cognitive tasks.

## 4  APPLICATION

### 4.1  PERCEPTUAL DENOSING

Perceptual denoising is critical for sensory processing, enabling the brain to filter out noise while preserving accurate representations of sensory inputs. To evaluate the denoising and maintenance capabilities of our Recurrent Divisive Normalization (RDN) model, we employed the Random Dot Kinematogram (RDK) task, a well-established paradigm for studying sensory perception under noisy conditions (Williams & Sekuler, 1984).

In the RDK task, a number of dots are randomly, with a subset moving coherently in a dominant direction (Fig. 3A). Task difficulty is modulated by coherence, defined as the percentage of dots moving towards the dominant direction. For this study, we simulated a 2-Alternative Foice Choice (2AFC) motion discrimination task (left vs. right) using two independent Gaussian distributions with identical variance ($\sigma^2$) but distinct means ($\mu_1, \mu_2$, Fig. 3C), where the separation between means was controlled by the coherence parameter:

$$\text{coherence} = \frac{|\mu_1 - \mu_2|}{\sigma} \tag{17}$$

We use RDN model with $N = 2$ for this task, where $R_L, R_R$ encode evidence for left/right motion respectively. The model parameters were set to $\tau = 50\text{ms}$, $\beta = 2$ and $\eta = 1$, with a numerical simulation time step of $dt = 0.1\text{ms}$. Random input signals were sampled at 100 Hz and simulated using zero-order hold interpolation. In our simulations, input signals represented motion strength with means of 0.52 (signal 1) and 0.48 (signal 2) and a shared variance of 0.17 (Fig. 3B, lower panel). Stimuli were presented for 2 s (onset: 300 ms). The model's readout signal (Fig. 3B, upper panel) exhibited noise reduction significantly compared to the raw inputs. Probability density analyses (Fig. 3C&D) revealed that the original signals were indistinguishable due to noise ($d' = 0.20$), whereas the readout signals showed clear separation ($d' = 1.67$). D-prime ($d'$) is a measure of sensitivity in signal detection theory, calculated as:

$$d' = \frac{\mu_S - \mu_N}{\sqrt{\frac{\sigma_S^2 + \sigma_N^2}{2}}}, \tag{18}$$

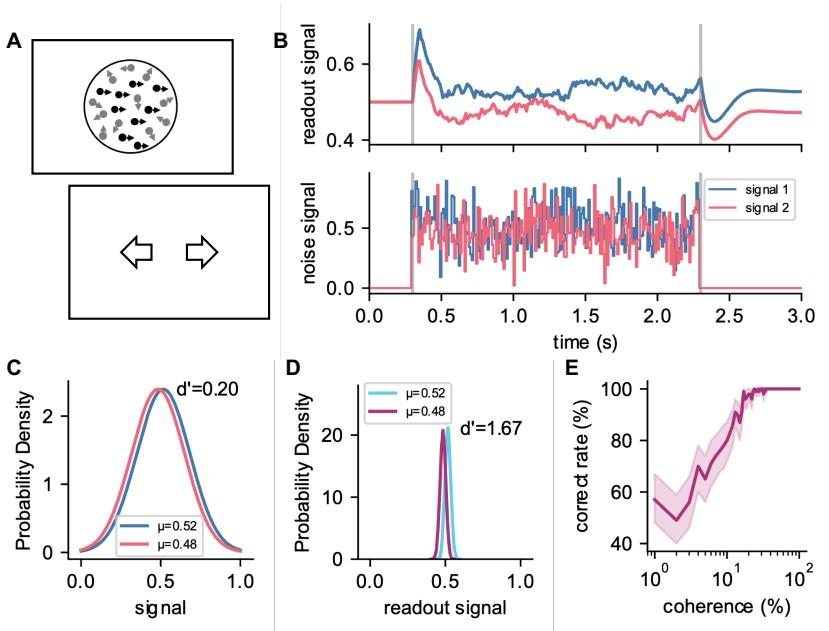

Figure 3: The Recurrent Divisive Normalization model exhibits noise reduction, making it useful in sensory perception task (RDK). **A**. Paradigm of the RDK task. A large number of randomly moving dots were presented with a dominant direction (left or right). The percentage of coherently moving dots (coherence, 0-100%) controls task difficulty. Lower coherence trials are more difficult with more noises. **B**. The model filters noise and gives denoised signals. Two inputs, "signal 1" and "signal 2" (lower panel), were used to simulate motion strength of the RDK task, corresponding to leftward and rightward motion of 10% coherence level. These signals are input to The stimuli was given at 300 ms and last for 2000 ms. Readout represents the model's state, showing the processed signals (upper panel). **C**. Probability density of the input signal, due to noise, the two signals are difficult to distinguish ($d' = 0.20$). **D**. Distribution of the "readout" signal from the model, showing much reduced noises. The processed signals can be better distinguished compared to the input signal($d' = 1.67$). **E**. Relationship between coherence of the RDK task and the accuracy of a simple discriminator using the processed signal (denoised representations) for choice. Each coherence level is sampled 100 times. Error bars represent the 95% confidence intervals (CIs).

where $\mu_S$ and $\mu_N$ are the means of the signal and noise, $\sigma_S^2$ and $\sigma_N^2$ are their variances.

The readout maintained stable mean values with reduced variance, demonstrating robust noise suppression. Moreover, the model sustained its representation after stimulus offset (Fig. 3B), highlighting its capacity of persistence memory.

To quantify representational accuracy across coherence levels, we simulated 100 coherence levels (1-100%). The readout state was assessed 1 s post-stimulus, with correctness determined by alignment to the ground-truth distribution (100 samples/condition). Accuracy increased monotonically with coherence, reaching 100% at 20% coherence (Fig.3E), consistent with primate neurophysiological data (Britten et al. (1992); Wang (2002), e.g., MT/LIP activity).

In summary, the RDN model achieves effective noise filtering and stable representation maintenance through divisive normalization and $\beta$-mediated feedback. Its performance aligns with psychophysical curves observed in primate dorsal stream areas, underscoring its utility for sensory decision-making under uncertainty.

## 4.2 PROBABILISTIC INFERENCE

Probabilistic inference is fundamental to adaptive behavior, particularly in tasks requiring flexible rule learning and switching (Behrens et al., 2007; 2008; Boorman et al., 2009). Cognitive flexibility

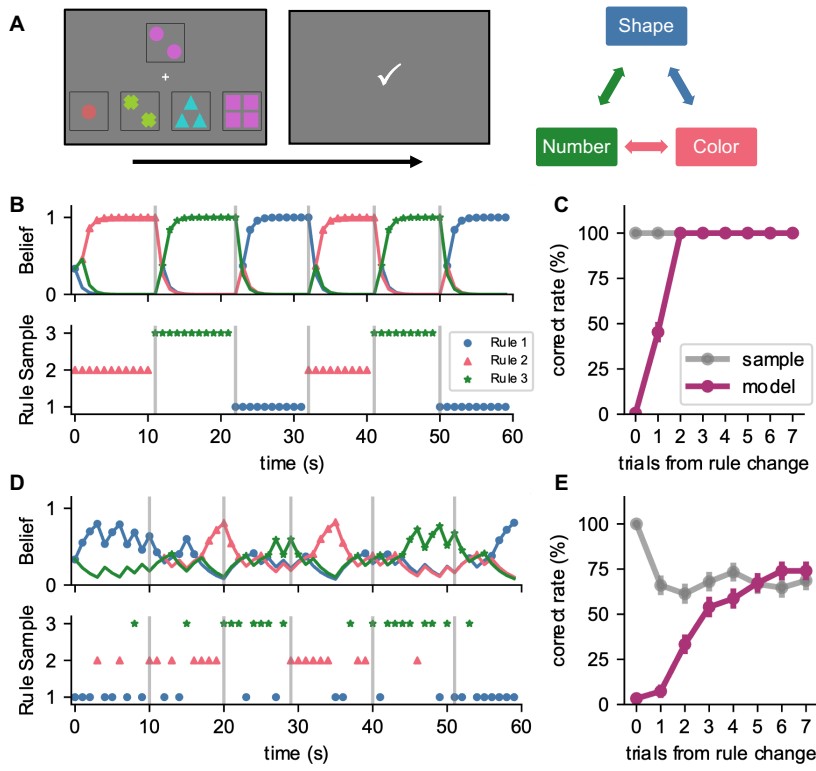

Figure 4: The Recurrent Divisive Normalization Model performs probabilistic inference in classical and probabilistic WCST task. **A**. The WCST task paradigm and its flexible changing structure: A reference card and some candidate cards are presented. Choices are evaluated based on matching shape (Rule 1), number (Rule 2), or color (Rule 3), with feedback indicating correctness or error according to the current rule, which should be used to learn the randomly switching rule. **B**. Rule design of a classical WCST task and the learned belief by the RDN model. Each block in the classic WCST employs a single, fixed rule for discrimination, comprising 9, 10, or 11 trials with a 1-second interval between trials. Different blocks are separated by gray vertical lines. Rule Sample refers to the rule used for discrimination, (Rule 1: blue circle, Rule 2: pink triangle, Rule 3: green pentagram). the model selects the rule with the highest belief as the basis for its choice in each trial (marked with the same symbols) and update belief based on feedback. **C**. Correct rate after change point show that the tested RDN model makes optimal rule learning and switching, i.e., win-stay, loss-shift. Gray circles represent simulated sampling data, purple circles indicate the model's accuracy. Error bars denote the standard error of the mean (SEM). **D**. Rule design of a probabilistic WCST (pWCST) task and model performance in the task. Each block employs probabilistic rules for discrimination, with a dominant rule having a probability of 0.7, making it impossible to relay on single feedback. The RDN model can learn and track the rule switch very well. Notations are consistent with B. **E**. Correct rate after probability change point show that the tested RDN model works well on the pWCST task. Note that to ensure accurate switch timing, the first and last trial of each block are fixed to the dominant rule. Notations are consistent with C.

are often measured using tasks like the Wisconsin Card Sorting Test (WCST) (Stemme et al., 2007), which can be modeled as probabilistic inference (D'Alessandro et al., 2020) or flexible gating (Liu & Wang, 2024). To evaluate the RDN model's capacity for belief updating, we examined its performance in both the classical and probabilistic versions of WCST. The classical WCST task assesses executive functions, including cognitive flexibility and attentional control. Participants match candidate cards to a reference card based on a latent rule (e.g., shape number or color; Fig. 4A). Feedback indicates correctness, and rules switch pseudorandomly, demanding rapid adaptation.

In our simulations (Fig. 4B), the RDN model comprised three excitatory units, encoding the belief of the corresponding rules. We use $\tau = 50$ms, $\beta = 2$ and $\eta = 1$ for this task, and $dt = 0.1$ms for

numerical simulations. Feedback inputs were set to 1 for unit corresponding to chosen rule and 0 for other units at correct trials, otherwise 0 for the chosen rule and 0.5 for others. Feedback inputs lasted for 50 ms per trial. Rules persisted for 9–11 trials before switching. The model achieved rapid rule-switching detection within two trials (Fig. 4B&C), consistent with theoretically optimal strategy bounds. Analysis of 150 rule blocks (Fig. 4C) confirmed this behavior, demonstrating the model's ability to leverage feedback for exact belief updating.

The probabilistic WCST task introduces greater complexity: feedback follows the dominant rule probabilistically (70%), occasionally adheres to incorrect rules (30%). This requires integration of historical reward to mitigate stochastic feedback, rather than adopt the "win-stay, loss-shift" strategy. With $\tau$ adjusted to 50 ms (other parameters unchanged), the model maintained robust performance (Fig. 4D&E). With a 150-block experiment, the accuracy from rule change point stabilized near the probability of the dominant rule(0.7), reflecting effective longer-term reward integration despite interference.

The RDN model achieves rapid rule switching in deterministic contexts and sustains performance under probabilistic feedback, mirroring human adaptive learning. Its success in both WCST variants underscores its utility for modeling approximate Bayesian inference in dynamic environments (Behrens et al., 2007).

## 5 DISCUSSION AND CONCLUSION

Our study presents a unified cortical circuit model that integrates divisive normalization with self-excitation to achieve both robust sensory processing and stable memory maintenance. This work bridges two fundamental but traditionally separate lines of research: normalization as a canonical cortical operation for noise-resistant encoding (Carandini & Heeger, 2011), and attractor dynamics for persistent information storage (Compte et al., 2000; Wang, 2002). The model's mathematical framework demonstrates how these mechanisms coexist within a single microcircuit, offering a parsimonious alternative to modular architectures requiring specialized subsystems. Functionally, it replicates both noise-robust sensory encoding (Britten et al., 1992) and cognitive flexibility in rule-switching tasks (Liu & Wang, 2024), suggesting common computational principles may underlie diverse neural functions.

The model resolves a critical theoretical gap. While divisive normalization has been formalized as a fundamental nonlinear operation (Kouh & Poggio, 2008), and attractor networks are well-established for memory maintenance (Constantinidis & Klingberg, 2016), their integration remained unexplored. Our framework reveals that normalization not only suppresses noise but also stabilizes attractor dynamics against input fluctuations—a prediction testable through combined electrophysiology and perturbation experiments. While aligning with the emerging concept of cortical "canonical computations" (Heeger, 2017) our work uniquely demonstrates how these computations synergize within a minimal circuit architecture, while may represent a plausible neural-circuit implementation of the Bayesian brain hypothesis (Knill & Pouget, 2004; Friston, 2009).

Several limitations highlight future directions. First, while the model captures core phenomena, predictions like persistent normalization during memory delays await experimental validation—a challenge shared by many theoretical studies (Sreenivasan et al., 2014). Second, rate-based dynamics simplify biological details such as spiking neurons and dendritic computations (Larkum et al., 2022). Third, pre-tuned weights omit developmental plasticity, though this simplification is common in foundational work (Stemme et al., 2007).

Future research should pursue: Experimental tests of neural signatures (e.g., normalized firing in prefrontal memory tasks, brain region specific parameters emphasis different functions); Introduce nonlinear activation functions to enable the model to exhibit richer dynamic properties and perform more cognitive tasks; Spiking implementations to assess temporal precision; Plasticity mechanisms for self-organized circuit tuning.

By unifying noise suppression and memory maintenance in a single architecture, this work challenges modular brain views and offers new principles for bio-inspired AI. The demonstrated synergy of canonical computations opens avenues to understand cortical efficiency and design adaptive neural systems.

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

# A APPENDIX

## A.1 EIGENVALUES OF A TYPE OF ARROWHEAD MATRIX

**Theorem 1.** *Consider the following $N + 1$ dimensional arrowhead matrix:*

$$\boldsymbol{M} = \begin{bmatrix} a\boldsymbol{I}_N & \boldsymbol{v} \\ \boldsymbol{w}^\top & b \end{bmatrix}, \tag{19}$$

*where $a, b \in R$ are scalars, $\boldsymbol{I}_N$ is an $N$-th order identity matrix, and $\boldsymbol{v}, \boldsymbol{w}$ are $N$-dimensional vectors, $\boldsymbol{w} \neq \boldsymbol{0}$. The eigenvalues of matrix $\boldsymbol{M}$ are given explicitly by:*

1. *$\lambda = a$ with multiplicity $N - 1$;*

2. *Two distinct eigenvalues:*

$$\lambda_\pm = \frac{(a + b) \pm \sqrt{(a - b)^2 + 4\boldsymbol{w}^\top \boldsymbol{v}}}{2}. \tag{20}$$

*which are real if $(a - b)^2 + 4\boldsymbol{w}^\top \boldsymbol{v} \geq 0$, and form a complex conjugate pair otherwise.*

*Proof.* Define a $(N - 1)$-dimensional subspace

$$\mathcal{V}_\perp = \{\boldsymbol{u} \in \mathbb{R}^n \mid \boldsymbol{w}^\top \boldsymbol{u} = 0\} \tag{21}$$

For any $\boldsymbol{u} \in \mathcal{V}_\perp$, the vector $[\boldsymbol{u}, 0]^\top$ satisfies

$$\boldsymbol{J} \begin{bmatrix} \boldsymbol{u} \\ 0 \end{bmatrix} = \begin{bmatrix} a\boldsymbol{I}_N \boldsymbol{u} + \boldsymbol{v} \cdot 0 \\ \boldsymbol{w}^\top \boldsymbol{u} + b \cdot 0 \end{bmatrix} = a \begin{bmatrix} \boldsymbol{u} \\ 0 \end{bmatrix}$$

which refers to $(N - 1)$-fold eigenvalues corresponding to the eigenvector $[\boldsymbol{u}, 0]^\top$.

Let the remaining eigenvectors have the form $[m\boldsymbol{v}, n]^T$, with $m, n \in \mathbb{R}$ and substitute it into the eigen equation, we have

$$\begin{cases} ma\boldsymbol{v} + n\boldsymbol{v} = \lambda m\boldsymbol{v} \\ m\boldsymbol{w}^\top \boldsymbol{v} + bn = \lambda n. \end{cases}$$

Simplify to obtain

$$n = m(\lambda - a),$$
$$m(\boldsymbol{w}^\top \boldsymbol{v}) + n(b - \lambda) = 0$$

and after eliminating $n$, we obtain

$$m(\boldsymbol{w}^\top \boldsymbol{v}) + m(\lambda - a)(b - \lambda) = 0$$

which can be decompsed into a trivial equation $m = 0$ and equation

$$(\lambda - a)(\lambda - b) - \boldsymbol{w}^\top \boldsymbol{v} = 0 \tag{22}$$

Solving this quadratic equation explicitly, we obtain the remaining two eigenvalues:

$$\lambda_\pm = \frac{(a + b) \pm \sqrt{(a - b)^2 + 4\boldsymbol{w}^\top \boldsymbol{v}}}{2}.$$

Note that the discriminant

$$\Delta = (a - b)^2 + 4\boldsymbol{w}^\top \boldsymbol{v}$$

may be negative. In that case, the two nontrivial eigenvalues $\lambda_\pm$ form a complex conjugate pair.

Therefore, the eigenvalues of $\boldsymbol{M}$ are fully characterized by:

$$\{\underbrace{a, \dots, a}_{N-1}, \lambda_+, \lambda_-\}.$$

This completes the proof. $\square$

## A.2 STABILITY OF THE SIMPLIFIED MODEL WITH INPUTS

**Theorem 2.** *Given dynamic system defined with Eqs. (12–13), For any parameters $\tau > 0$, $\beta > 0$, $\eta > 0$ and input $I_i > 0$, the unique steady-state $(\boldsymbol{R^*}, G^*)$ is an attractor.*

*Proof.* From Eq. (6–7), the eigenvalues of the system is given by

$$\lambda_i = a = \frac{-1 + \beta/(\eta + G^*)}{\tau}, \quad (i = 1, \dots, N-1), \tag{23}$$

$$\lambda_+ = \frac{(a - 1/\tau) + \sqrt{(a + 1/\tau)^2 - 4G^*/(\eta + G^*)/\tau^2}}{2}, \tag{24}$$

$$\lambda_- = \frac{(a - 1/\tau) - \sqrt{(a + 1/\tau)^2 - 4G^*/(\eta + G^*)/\tau^2}}{2}, \tag{25}$$

It is obviously, $\lambda_i = a < 0$ and $\Re(\lambda_-) < 0$ if $\beta < \eta + G^*$, while from Eq. equation 4, we have

$$
\begin{aligned}
G^* + \eta - \beta &= \frac{-(\eta - \beta) + \sqrt{(\eta - \beta)^2 + 4T}}{2} + \eta - \beta \\
&= \frac{(\eta - \beta) + \sqrt{(\eta - \beta)^2 + 4T}}{2} \\
&> \frac{(\eta - \beta) + |\eta - \beta|}{2} \geq 0,
\end{aligned}
\tag{26}
$$

always hold for $T > 0$. To proof $\Re(\lambda_+) < 0$, we simplify Eq. equation 24 as

$$
\begin{aligned}
2\lambda_+ &= a - \frac{1}{\tau} + \sqrt{(a + \frac{1}{\tau})^2 - \frac{4G^*}{\tau^2(\eta + G^*)}} \\
&= -\frac{2}{\tau} + \frac{\beta}{\tau(\eta + G^*)} + \sqrt{\frac{\beta^2}{\tau^2(\eta + G^*)^2} - \frac{4G}{\tau^2(\eta + G^*)}} \\
&= \frac{-2(\eta + G^*) + \beta}{\tau(\eta + G^*)} + \frac{\sqrt{\beta^2 - 4G^*(\eta + G^*)}}{\tau(\eta + G^*)}
\end{aligned}
$$

Note that if $\beta^2 - 4G^*(\eta + G^*) < 0$, the real part

$$\Re(\lambda_+) = \frac{-2(\eta + G^*) + \beta}{\tau(\eta + G^*)} = \frac{-2(\eta - \beta + G^*) - \beta}{\tau(\eta + G^*)} < 0.$$

If $\beta^2 - 4G^*(\eta + G^*) > 0$, we only need to proof

$$
\begin{aligned}
-2(\eta + G^*) + \beta + \sqrt{\beta^2 - 4G^*(\eta + G^*)} &< 0 \\
\sqrt{\beta^2 - 4G^*(\eta + G^*)} &< 2(\eta + G^*) - \beta \\
\beta^2 - 4G^*(\eta + G^*) &< (2(\eta + G^*) - \beta)^2 \\
\beta^2 - 4G^*(\eta + G^*) &< 4(\eta + G^*)^2 - 4\beta(\eta + G^*) + \beta^2 \\
-4G^*(\eta + G^*) &< 4(\eta + G^*)^2 - 4\beta(\eta + G^*) \\
-G^* &< (\eta + G^*) - \beta,
\end{aligned}
$$

which also be satisfied. Therefore, all eigenvalues have a negative real part, meaning the steady-state $(\boldsymbol{R^*}, G^*)$ is an attractor.

This completes the proof. □

## A.3 FIXED POINT PROPERTIES

With $I_i = 0$, denote $S = \sum_{i=1}^{N} w_i R_i, \rho_i = \frac{w_i R_i}{S}$, which satisfies $\sum_{i=1}^{N} \rho_i = 1$, and we have

$$
\begin{aligned}
\tau \frac{\mathrm{d}\rho_i}{\mathrm{d}t} &= w_i \frac{\tau}{S} \frac{\mathrm{d}R_i}{\mathrm{d}t} - w_i \frac{\tau R_i}{S^2} \frac{\mathrm{d}S}{\mathrm{d}t} \\
&= w_i \frac{\tau}{S} \frac{1}{\tau} \left( -R_i + \frac{\beta R_i}{\eta + G} \right) - w_i \frac{\tau R_i}{S^2} \sum_{j=1}^{n} w_j \frac{\mathrm{d}R_j}{\mathrm{d}t} \\
&= w_i \frac{\tau}{S} \frac{1}{\tau} \left( -R_i + \frac{\beta R_i}{\eta + G} \right) - w_i \frac{\tau R_i}{S^2} \frac{1}{\tau} \sum_{j=1}^{n} w_j \left( -R_j + \frac{\beta R_j}{\eta + G} \right) \\
&= w_i \frac{\tau}{S} \frac{1}{\tau} \left( -R_i + \frac{\beta R_i}{\eta + G} \right) - w_i \frac{\tau R_i}{S^2} \frac{1}{\tau} \left( -S + \frac{\beta S}{\eta + G} \right) \\
&= 0
\end{aligned} \tag{27}
$$

With the inputs exerted to the system, we have

$$
\begin{aligned}
\tau \frac{\mathrm{d}\rho_i}{\mathrm{d}t} &= w_i \frac{\tau}{S} \frac{\mathrm{d}R_i}{\mathrm{d}t} - w_i \frac{\tau R_i}{S^2} \frac{\mathrm{d}S}{\mathrm{d}t} \\
&= w_i \frac{\tau}{S} \frac{1}{\tau} \left( -R_i + \frac{I_i + \beta R_i}{\eta + G} \right) - w_i \frac{\tau R_i}{S^2} \sum_{j=1}^{n} w_j \frac{\mathrm{d}R_j}{\mathrm{d}t} \\
&= w_i \frac{\tau}{S} \frac{1}{\tau} \left( -R_i + \frac{I_i + \beta R_i}{\eta + G} \right) - w_i \frac{\tau R_i}{S^2} \frac{1}{\tau} \sum_{j=1}^{n} w_j \left( -R_j + \frac{I_j + \beta R_j}{\eta + G} \right) \\
&= w_i \frac{1}{S} \left( -R_i + \frac{I_i + \beta R_i}{\eta + G} \right) - w_i \frac{R_i}{S^2} \left( -S + \frac{\beta S}{\eta + G} + \sum_{j=1}^{n} \frac{w_j I_j}{\eta + G} \right) \\
&= w_i \left( \frac{I_i}{S(\eta + G)} - \frac{R_i}{S^2(\eta + G)} \sum_{j=1}^{n} w_j I_j \right)
\end{aligned} \tag{28}
$$

When the system has converged to the fixed point, we have $I_i = R_i^*(\eta + G^* - \beta)$, and

$$
\begin{aligned}
\tau \frac{\mathrm{d}\rho_i}{\mathrm{d}t} &= w_i \left( \frac{I_i}{S(\eta + G^*)} - \frac{R_i^*}{S^2(\eta + G^*)} \sum_{j=1}^{n} w_j I_j \right) \\
&= w_i \left( \frac{R_i^*(\eta + G^* - \beta)}{S(\eta + G^*)} - \frac{R_i^*(\eta + G^* - \beta) \sum_{j=1}^{n} w_j R_j^*}{S^2(\eta + G^*)} \right) \\
&= w_i \left( \frac{R_i^*(\eta + G^* - \beta)}{S(\eta + G^*)} - \frac{R_i^*(\eta + G^* - \beta)S}{S^2(\eta + G^*)} \right) \\
&= 0
\end{aligned} \tag{29}
$$

It means that with inputs exerted to the system, the balances among proportions $\rho_i$ will initially be broken, in other words, the previous *memory* of the system will be *destroyed*, and $\rho_i$ will return to constant when the system converges to a new fixed point, forming a new representation of inputs.

