# OpenReview forum: "A Unified Cortical Circuit Model with Divisive Normalization and Self-Excitation for Robust Representation and Memory Maintenance"
_ICLR.cc/2026/Conference — Submitted to ICLR 2026_

### Official Review · Reviewer_3z9q · 2025-10-29

**Soundness:** 2
**Presentation:** 3
**Contribution:** 2
**Rating:** 4
**Confidence:** 3

**Summary:**

The paper proposes a recurrent neural circuit model that integrates divisive normalization and self-excitation to jointly achieve robust sensory encoding and stable memory maintenance. The authors analytically show that, under appropriate parameters, the system forms a continuous attractor capable of denoising noisy inputs and sustaining representations after stimulus removal. Through mathematical derivations and two illustrative tasks—RDK (for perceptual denoising) and WCST (for flexible rule inference)—the model demonstrates that a single cortical microcircuit can perform both noise-suppression and working-memory functions. The authors argue that this unified framework bridges two canonical cortical computations and could inform the design of biologically plausible artificial neural networks.

**Strengths:**

- The paper is clearly written. The motivation, model architecture, and experimental setup are all easy to follow.

- The theoretical analysis is solid and seems correct. The derivations (steady states, bifurcation, stability) connect well with the model’s qualitative behavior.

- The two chosen tasks (RDK and WCST) nicely illustrate how the same circuit can support both noise-robust encoding and persistent activity.

- The figures are clear and effectively convey the model’s behavior.

- The paper is well-grounded in existing computational neuroscience literature, and the link to canonical cortical computations (normalization and attractor dynamics) is thoughtfully made.

**Weaknesses:**

- I’m not fully convinced by the motivation of a unifying model of perceptual denoising and  memory maintenance. One could in principle pick any two cognitive computations and design a model to achieve both — why these two in particular, beyond the fact that they’ve been studied separately?

- The AI relevance statements in the abstract and discussion feel overstated. The model is elegant but quite simple; the claims about implications for artificial intelligence seem premature.

- The experiments are purely conceptual and don’t connect to biological or empirical data. That makes the “unified cortical circuit” claim somewhat speculative.

- The “unification” mainly comes from adding recurrent excitation to a divisive normalization framework. It’s an interesting idea, but the novelty feels incremental.

- The probabilistic inference task (pWCST) is more of a rule-tracking demonstration than a true implementation of Bayesian belief updating. The link is more conceptual than mechanistic.

- Parameter choices seem somewhat arbitrary, and the paper doesn’t show how sensitive the results are to these parameters.

**Questions:**

- In the experiments, how are the R–G weights determined? Are they always set to 1, or adjusted per task?
- What happens if $\beta$ is close to $\eta$ in your experiments? Will the model still be able to keep memory?
- I'm not sure if I follow the experiment setup of the Bayesian inference experiment. For example, how are the (0/0.5/1) feedback inputs provided to the model?
- It is known that RNNs can perform perceptual denoising and keeping working memory (e.g., see refs below) - how could you relate your model to RNN models? Maybe worth mentioning it at least in the discussion.

References:
- Masse, Nicolas Y., et al. "Circuit mechanisms for the maintenance and manipulation of information in working memory." Nature neuroscience 22.7 (2019): 1159-1167.
- Song, H. Francis, Guangyu R. Yang, and Xiao-Jing Wang. "Training excitatory-inhibitory recurrent neural networks for cognitive tasks: a simple and flexible framework." PLoS computational biology 12.2 (2016): e1004792.

---

### Official Review · Reviewer_bExF · 2025-10-30

**Soundness:** 2
**Presentation:** 3
**Contribution:** 3
**Rating:** 4
**Confidence:** 4

**Summary:**

The paper introduces a simple model that combines self‑excitated excitatory unit and shared inhibitory neuron. The authors show analytically that the system has a unique steady-state solution and can exhibit a continuous attractor when the self-excitation β exceeds the semi‑saturation constant η. They derive conditions for stability and show that with inputs the fixed point corresponds to normalized input ratios.

Also really simple, this model has really interesting properties.

The model is applied to a Random Dot Kinematogram denoising task and to both deterministic and probabilistic Wisconsin Card Sorting Tasks. In the RDK experiment, two excitatory units (representing left/right motion) with β = 2, η = 1 and τ = 50 ms denoise two noisy input streams; the authors report that the d′ improves from 0.20 to 1.67 and that the network retains the normalized ratio after the stimulus ends. In the WCST experiments, three units with the same parameters update beliefs about rules based on feedback and can switch rules within two trials or track probabilistic rules.

I am giving the paper an overall 4 "marginally below the acceptance threshold" but will consider increasing that grade if the weaknesses are adresses.

**Strengths:**

* Unified framework: The paper proposes a simple dynamical system that reduces to classical divisive normalization when β=0 and generalizes known recurrent normalization models for β=1, giving a nice mathematical link between normalization and attractor dynamics. The steady‑state derivation and identification of a transcritical bifurcation when β crosses η are clearly explained, and the continuous attractor analysis is analytically grounded.
* Clarity of writing and figures: The model and its dynamics are well illustrated (Fig. 1 and Fig. 2). The tasks are described clearly, and the simulation results are easy to follow. Discussion and future directions acknowledge limitations and suggest spiking extensions.

**Weaknesses:**

* Even when if goal is to illustrate a concept rather than optimize performance, you need to provide some justification or exploration of parameter choices because it speaks to the robustness and generality of the proposed mechanism. In your paper, the key results hinge on a few manually chosen values (for example, β = 2 and η = 1 with τ = 50 ms) in both tasks. Without showing how the system behaves when these parameters vary, readers cannot tell whether the ability to maintain normalized representations and perform inference is a general property of the architecture or a consequence of tuning to specific numbers.

* Besides hyper‑parameter tuning, the paper lacks quantitative comparisons with alternative models. For instance, it does not show how the RDN’s denoising compares with classical divisive normalization (β = 0), recurrent normalization (β = 1) or attractor networks on the same tasks.

**Questions:**

1. Can you provide baseline comparisons with β = 0 and β = 1, or with standard attractor/Bayesian models on your tasks?
2. How sensitive are the results to β, η and τ ?

---

### Official Review · Reviewer_QwCw · 2025-11-03

**Soundness:** 3
**Presentation:** 3
**Contribution:** 2
**Rating:** 4
**Confidence:** 4

**Summary:**

The paper proposes a cortical circuit model with divisive normalization and self-excitation, attempting to unify noise resistance and information maintenance. The circuit model looks like an extension based on (recurrent) divisive normalization models (ref. 24-27) with a major revision of inserting a new parameter $\beta$ representing self-excitation (Eq. 1). The key contribution claimed by the authors is probability demonstrating that high-dimensional continuous attractors can emerge in the proposed extremely minimal model from the dynamical system perspective. The paper further uses the same circuit model architecture but with different parameters to implement a sensory perception task (Fig. 3) and the Wisconsin card sorting task (Fig. 4).

**Strengths:**

- Analytical solutions of the nonlinear neural dynamics.

- Utilize the circuit to realize two cognitive tasks (Fig. 3 and 4)

**Weaknesses:**

### 1. The neural dynamics is over-simplified due to the lack of recurrent excitation.

### 2. The study doesn't fully utilize the analytical tractability of a minimal model to understand the computational and algorithmic mechanism of neural circuits.

The theoretical analysis of the circuit only focuses on the stability analysis of the simple model. In contrast, the comp-neuro field has developed more comprehensive theoretical analyses on more complex recurrent networks that include both recurrent connections across E neurons and static divisive normalizations. In addition, the recent development on Stabilized Supralinear Networks (SSN) by Ken Miller also yielded much deeper dynamical system understanding than the current paper, even if the SSN doesn't contain _explicit_ divisive normalization (but the supralinear inhibitory neurons activation function enables E neurons realize input-output curves similar to divisive normalization). Combined, I feel the stability analysis on the proposed minimal model doesn't provide a significant advance on circuit dynamical theories when compared with the papers mentioned above.

### 3.The disconnection of the theoretical analysis and the circuit mechanism of cognitive tasks in Figs. 3-4.
While I appreciate the theoretical analysis in Sec. 2, Sec. 4 only uses numerical simulations to show the circuit model can realize the two cognitive tasks, without utilizing the theory in Sec. 2 to gain deep insight into the coding/algorithmic mechanism in the circuit model underlying the two cognitive tasks. This would be a waste of the analytical results of the proposed nonlinear circuit dynamics, and that's why I think the work is still in the preliminary stage. Since we have analytical results, we could ask many deep questions. For example, how do the circuit parameters like $\beta$ and dynamic divisive normalization affect the coding performance in the Fig. 3 task? What is the underlying probabilistic inference algorithm of the circuit in performing Fig. 4 task (variational Bayes or sampling), and/or how is the belief represented by neuronal population activities (firing rate proportional to probability or log-probability, or sampling-based representation)? The absence of these insights significantly limits the depth of this paper. In addition, considering that the two cognitive tasks have been extensively studied in neuroscience research, the paper doesn’t provide new (theoretical) insight into the circuit mechanism, nor a comprehensive comparison with existing studies of circuit mechanisms underlying the two cognitive tasks.

**Questions:**

I have no detailed question on the study. I would like to disclose that I also reviewed this paper submitted to NeurIPS 2025 earlier and accessed all questions raised by all reviewers. After browsing the ICLR version, it seems that there is no significant change of the text, even in the Discussion section.

---

### Official Review · Reviewer_Lywm · 2025-11-04

**Soundness:** 2
**Presentation:** 2
**Contribution:** 2
**Rating:** 6
**Confidence:** 3

**Summary:**

The paper here proposes the Recurrent Divisive Normalization (RDN) circuit model where the purpose is to bridge the gap between denoising capability of neural coding and the ability to maintain sustained representations in these neural coding, considering that these problems are considered to be of separate domains. The author shows that under certain parameter conditions, the model can show normalized representation of the input while also capable of maintaining persistent representation of the input when the input is withdrawn. The author performed a model analysis to show the stability of the model with/without input hence showing the model maintaining persistent representation. The author performed random dot kinematogram (RDK) task, and on both classical and probabilistic versions of Wisconsin Card Sorting Test (WCST) to show the model’s robustness with noise.

**Strengths:**

It is interesting that the author showcases a model that is capable of being both an attractor network and a normalisation model, hence capable of sustaining persistent activities of representation while also being capable of denoising in a specific regime. Moreover, it is nice to see that there is some analytical support done for the persistent representations, as well as cross-demain demonstrations of the models’ normalization.

**Weaknesses:**

I do feel like the abstract is lacking in describing what the key contribution is. It feels lackluster and I can only understand this as I go further into the paper.
Grammar/ Structuring/ Spelling error needs to be fixed throughout. Please revise the paper for these kinds of errors. Some examples to check:
Line 70-71: “ but also forms a continuous attractor that persistently maintains those representations after input withdrawn.”
Line 48: “While there exists different perspectives”
Line 51-52: “Bayesian approach consider the neuron population”
Line 94: “β = 1 lead to a model”
Line 306: “2-Alternative Foice Choice”
Do define what and how the 3 excitatory units for WCST are done. It is pretty general in mentioning just shape, number and color.
Although these experiments are given, there are lots of other models that could be compared to this that haven’t been mentioned. E.g. attractor networks (either discrete, continuous), classic or soft winner take all models, etc. Could experiments to further analyse these differences be done for the model?
The number of classes (N) is small in the current experiments, i.e. RDK (N=2) and WCST (N=4). Could you evaluate larger N to assess scalability (e.g., 8, 16, 64) and whether denoising and persistence still hold?
Figure 2 is confusing for beta = eta. Although the black dot is supposed to be an unstable fixed point, this seems to be converging towards the black dot. Could you explain this part?

**Questions:**

Questions for the author has been addressed alongside the weakness in the weakness section. Please check the weakness section for this.

---

### Meta-Review · Area_Chair_tMrc · 2026-01-05

**Summary:**

The reviewers were on the fence about the submission (4, 4, 4, 6). The primary concerns were that the model is overly simplistic, but despite this simplicity the theoretical analysis is shallow. There were also concerns about the motivation and hyperparameter choices. Reviewer QwCw also stated that they reviewed this paper for NeurIPS 2025 and that the authors did not appear to address concerns that had been raised by reviewers during the NeurIPS reviewer period.

**Reviewer Concerns:**

The authors did not respond to the reviews.

**Reviewer Scores:**

Since the authors did not respond, I do not think the reviewers would have changed their scores.

---

### Decision · Program_Chairs · 2026-01-26

Reject